# Voxurf: Voxel-based Efficient and Accurate Neural Surface Reconstruction

**Tong Wu**[1,2], **Jiaqi Wang**[1✉], **Xingang Pan**[3], **Xudong Xu**[2], **Christian Theobalt**[3], **Ziwei Liu**[4], **Dahua Lin**[1,2,5✉]
[1]Shanghai AI Laboratory, [2]The Chinese University of Hong Kong, [3]Max Planck Institute for Informatics,
[4]S-Lab, Nanyang Technological University, [5]Centre of Perceptual and Interactive Intelligence
{wt020, xx018, dhlin}@ie.cuhk.edu.hk, wangjiaqi@pjlab.org.cn,
{xpan,theobalt}@mpi-inf.mpg.de, ziwei.liu@ntu.edu.sg

## Abstract

Neural surface reconstruction aims to reconstruct accurate 3D surfaces based on multi-view images. Previous methods based on neural volume rendering mostly train a fully implicit model with MLPs, which typically require hours of training for a single scene. Recent efforts explore the explicit volumetric representation to accelerate the optimization via memorizing significant information with learnable voxel grids. However, existing voxel-based methods often struggle in reconstructing fine-grained geometry, even when combined with an SDF-based volume rendering scheme. We reveal that this is because 1) the voxel grids tend to break the color-geometry dependency that facilitates fine-geometry learning, and 2) the under-constrained voxel grids lack spatial coherence and are vulnerable to local minima. In this work, we present **Voxurf**, a voxel-based surface reconstruction approach that is both efficient and accurate. Voxurf addresses the aforementioned issues via several key designs, including 1) a two-stage training procedure that attains a coherent coarse shape and recovers fine details successively, 2) a *dual color network* that maintains color-geometry dependency, and 3) a *hierarchical geometry feature* to encourage information propagation across voxels. Extensive experiments show that Voxurf achieves high efficiency and high quality at the same time. On the DTU benchmark, Voxurf achieves higher reconstruction quality with a 20x training speedup compared to previous fully implicit methods.

## 1 Introduction

Neural surface reconstruction based on multi-view images has recently seen dramatic progress. Inspired by the success of Neural Radiance Fields (NeRF) (Mildenhall et al., 2020) on Novel View Synthesis (NVS), recent works follow the neural volume rendering scheme to represent the 3D geometry with a signed distance function (SDF) or occupancy field via a fully implicit model (Oechsle et al., 2021; Yariv et al., 2021; Wang et al., 2021). These approaches train a deep multilayer perceptron (MLP), which takes in hundreds of sampled points on each camera ray and outputs the corresponding color and geometry information. Pixel-wise supervision is then applied by measuring the difference between the accumulated color on each ray and the ground truth. Struggling with learning all the geometric and color details with a pure MLP-based framework, these methods require hours of training for a single scene, which substantially limits their real-world applications.

Recent advances in NeRF accelerate the training process with the aid of an explicit volumetric representation (Sun et al., 2022a; Yu et al., 2022; Chen et al., 2022). These works directly store and optimize the geometry and color information via explicit voxel grids. For example, the density of a queried point can be readily interpolated from the eight neighboring points, and the view-dependent color is either represented with spherical harmonic coefficients (Yu et al., 2022) or predicted by shallow MLPs that take learnable grid features as auxiliary inputs (Sun et al., 2022a). These approaches achieve competitive rendering performance at a much lower training cost ($< 20$ minutes). However, their 3D surface reconstruction results cannot faithfully represent the exact geometry, suffering from

---

✉Corresponding authors. Our code is available at https://github.com/wutong16/Voxurf.

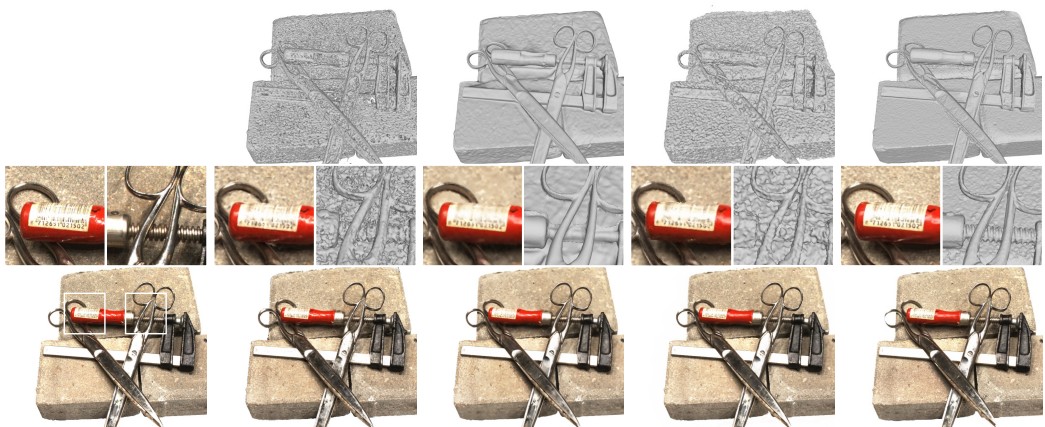

Ground Truth      (a) DVGO at 4 mins.     (b) NeuS at 5.5 hours.    (c) DVGO+NeuS at 12 mins.    (d) Ours at 15 mins.

Figure 1: **Comparisons among different methods for surface reconstruction and novel view synthesis.**
**(a)** DVGO (v2) (Sun et al., 2022a;b) benefits from the fastest convergence but suffers from a poor surface; **(b)** NeuS (Wang et al., 2021) produces decent surfaces after a long training time, while high-frequency details are lost in both the geometry and the image; **(c)** the straightforward combination of DVGO and NeuS produces continuous but noisy surfaces; **(d)** our method achieves around 20x speedup than NeuS and recovers high-quality surfaces and images with fine details. All the training times are tested on a single Nvidia A100 GPU.

conspicuous noise and holes (Fig. 1 (a)). It is due to the inherent ambiguity of the density-based volume rendering scheme, and the explicit volumetric representation introduces additional challenges.

In this work, we aim to take advantage of the explicit volumetric representation for efficient training and propose customized designs to harvest high-quality surface reconstruction. A straightforward idea for this purpose is to embed the SDF-based volume rendering scheme (Wang et al., 2021; Yariv et al., 2021) into explicit volumetric representation frameworks (Sun et al., 2022a). However, we find this naïve baseline model not working well by losing most of the geometry details and producing undesired noise (Fig. 1 (c)). We reveal several critical issues for this framework as follows. First, in fully implicit models, the color network takes surface normals as inputs, effectively building *color-geometry dependency* that facilitates fine-geometry learning. However, in the baseline model, the color network tends to depend more on the additional under-constrained voxel feature grid input, thus breaking color-geometry dependency. Second, due to a high degree of freedom in optimizing a voxel grid, it is hard to maintain a globally coherent shape without additional constraints. Individual optimization for each voxel point hinders the information sharing across the voxel grid, which hurts the surface smoothness and introduces local minima. We'll unveil these effects and introduce the insight for our architecture design via an empirical study in Sec. 4

To tackle the challenges, we introduce **Voxurf**, an efficient pipeline for accurate **Vox**el-based **surf**ace reconstruction: 1) We leverage a two-stage training process that attains a coherent coarse shape and recovers fine details successively. 2) We design a *dual color network* that is capable of representing a complex color field via a voxel grid and preserving the color-geometry dependency with two sub-networks that work in synergy. 3) We also propose a *hierarchical geometry feature* based on the SDF voxel grid to encourage information sharing in a larger region for stable optimization. 4) We introduce several effective regularization terms to boost smoothness and reduce noise.

We conduct experiments on the DTU (Jensen et al., 2014) and BlendedMVS (Yao et al., 2020) datasets for quantitative and qualitative evaluations. Experimental results demonstrate that Voxurf achieves lower Chamfer Distance on the DTU (Jensen et al., 2014) benchmark than a competitive fully implicit method NeuS (Wang et al., 2021) with around 20x speedup. It also achieves remarkable results on the auxiliary task of NVS. As illustrated in Fig. 1, our method is shown to be superior in preserving high-frequency details in both geometry reconstruction and image rendering compared to the previous approaches. In summary, our contributions are highlighted below:

1. Our approach enables around 20x speedup for training compared to the SOTA methods, reducing the training time from over 5 hours to 15 minutes on a single Nvidia A100 GPU.

2. Our approach achieves higher surface reconstruction fidelity and novel view synthesis quality, which is superior in representing fine details for both surface recovery and image rendering compared to previous methods.

3. Our study provides insightful observations and analysis of the architecture design of the explicit volumetric representation framework for surface reconstruction.

## 2 RELATED WORKS

**Multi-view 3D reconstruction**   Recently, implicit representations that encode the geometry and appearance of a 3D scene by neural networks have gained attention (Park et al., 2019; Chen & Zhang, 2019; Lombardi et al., 2019; Mescheder et al., 2019; Sitzmann et al., 2019b; Saito et al., 2019; Atzmon et al., 2019; Jiang et al., 2020; Zhang et al., 2021; Toussaint et al., 2022). Among them, a plethora of papers have explored neural surface reconstruction from multi-view images. Methods based on surface rendering (Niemeyer et al., 2020; Yariv et al., 2020; Liu et al., 2020b; Kellnhofer et al., 2021) regard the color of an intersection point of the ray and the surface as the final rendered color, while requiring accurate object masks and careful weight initialization. Recent approaches (Wang et al., 2021; Yariv et al., 2021; Oechsle et al., 2021; Darmon et al., 2022; Zhang et al., 2022; Liu et al., 2020a; Sitzmann et al., 2019a) based on volume rendering (Max, 1995) formulate the radiance fields and implicit surface representations in a unified model, thereby achieving the merits of both techniques. However, encoding the whole scene in pure MLP networks requires a long training time. In a departure from these works, we leverage learnable voxel grids and shallow color networks for quick convergence, as well as pursue fine details in surfaces and rendered images.

**Explicit volumetric representation**   Despite the success of implicit neural representations in 3D modeling, recent advances have integrated explicit 3D representations, *e.g.*, point clouds, voxels, and MPIs (Mildenhall et al., 2019), and received growing attention (Wizadwongsa et al., 2021; Xu et al., 2022; Lombardi et al., 2019; Wang et al., 2022; Fang et al., 2022). Instant-ngp (Müller et al., 2022) uses multi-resolution hashing for efficient encoding and implements fully-fused CUDA kernels. Plenoxels (Yu et al., 2022) represent a scene as a sparse 3D grid with spherical harmonics and are optimized two orders of magnitude faster than NeRF (Mildenhall et al., 2020). TensoRF (Chen et al., 2022) considers the full volume field as a 4D tensor and factorizes it into multiple compact low-rank tensor components for efficiency. The method most related to ours is DVGO (Sun et al., 2022a), which adopts a hybrid architecture design including voxel grids and a shallow MLP. Despite their remarkable results on novel view synthesis, none of them is designed to faithfully reconstruct the geometry of the scene. In contrast, we target at not only rendering photo-realistic images from novel viewpoints but also reconstructing high-quality surfaces with fine details.

## 3 PRELIMINARIES

**Volume rendering with SDF representation.** NeuS (Wang et al., 2021) represents a scene as an implicit SDF field parameterized by an MLP. The ray emitting from the camera center $o$ through an image pixel in the viewing direction $v$ can be expressed as $\{p(t) = o + tv | t \geq 0\}$. The rendered color for the image pixel is integrated along the ray with volume rendering (Max, 1995), which is approximated by $N$ discrete sampled points $\{p_i = o + t_i v | i = 1, ..., N, t_i < t_{i+1}\}$ on the ray:

$$\hat{C}(r) = \sum_{i=1}^{N} T_i \alpha_i c_i, \ \ T_i = \prod_{j=1}^{i-1} (1 - \alpha_j), \tag{1}$$

where $\alpha_i$ is the opacity value, and $T_i$ is the accumulated transmittance. The key difference between NeuS and NeRF is the formula of $\alpha_i$. In NeuS, $\alpha_i$ is formulated as:

$$\alpha_i = \max \left( \frac{\Phi_s(f(p(t_i))) - \Phi_s(f(p(t_{i+1})))}{\Phi_s(f(p(t_i)))}, 0 \right). \tag{2}$$

Here, $f(x)$ is the SDF function, and $\Phi_s(x) = (1 + e^{-sx})^{-1}$ is the Sigmoid function, where the $s$ value is learned or manually updated during training.

**Explicit volumetric representation.** DVGO (Sun et al., 2022a) represents the geometry with explicit density voxel grids $V^{(density)} \in \mathbb{R}^{1 \times N_x \times N_y \times N_z}$. It applies a hybrid architecture for color prediction that comprises a shallow MLP parameterized by $\Theta$ and a feature voxel grid $V^{(feat)} \in \mathbb{R}^{C \times N_x \times N_y \times N_z}$. Given a 3D position $p$ and the viewing direction $v$, the volume density $\sigma$ and color $c$ are estimated with:

$$\sigma = \text{interp}(p, V^{(density)}), \tag{3}$$

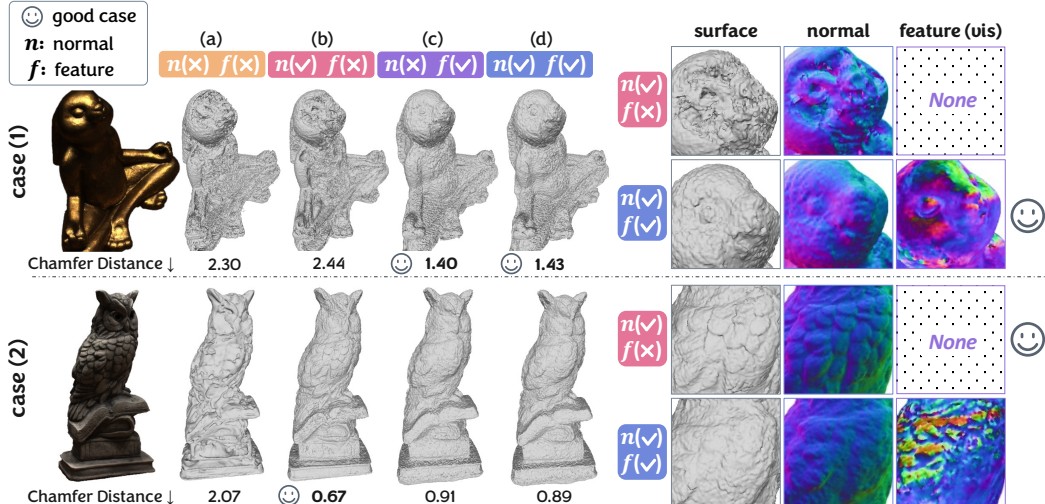

Figure 2: **Reconstruction results from different architecture designs.** The *surface normal* $n$ and *learnable feature* $f$ are both optional inputs to the color network. We show results of two cases under four settings on the **left**, and we zoom in to analyze the surfaces, normal fields, and feature fields on the **right**. Case (1) (a, c) and (b, d) show that the feature $f$ helps maintain a coherent shape, while case (2) (b, d) reveal that it discourages the reconstruction of geometry details since it disturbs the color-geometry dependency built by the normal $n$.

$$c = \text{MLP}_\Theta(\text{interp}(p, V^{(feat)}), p, v), \tag{4}$$

where 'interp' denotes the trilinear interpolation. Following NeRF (Mildenhall et al., 2020; Tancik et al., 2020), the positional encoding for both $p$ and $v$ is applied in Eqn. 4.

**Naïve Combination.** A straightforward combination of the two techniques is to replace the volume rendering in DVGO with the SDF-based volume rendering scheme as in Eqn. 1 and Eqn. 2. It serves as the naïve baseline in this work, which can hardly produce satisfactory results, as shown in Fig. 1 (c). We will cast light on this phenomenon via an empirical study in the next section.

## 4 STUDY ON ARCHITECTURE DESIGN FOR GEOMETRY LEARNING

In this section, we carry out some prior experiments with variants of the baseline model, aiming to figure out the key factors for architecture design in this task. Specifically, we employ an SDF voxel grid $V^{(sdf)}$ and apply Eqn. 2 for $\alpha$ calculation with a manually defined schedule for $s$. We start with a shallow MLP as the color network, where 1) the local feature $f$ interpolated from $V^{(feat)}$ and 2) the normal vector $n$ calculated by $V^{(sdf)}$ are both optional inputs. A decent surface reconstruction is expected to possess *a coherent coarse structure*, *accurate fine details*, and *a smooth surface*. We will next focus on these factors and analyze the effects of different architecture designs.

**The key to maintaining a coherent coarse shape.** Intuitively, the capacity of a shallow MLP is limited, and it can hardly represent a complex scene with different materials, high-frequency textures, and view-dependent lighting information. When the ground truth image encounters a rapid color-shifting, the volume rendering integration over an under-fitted color field results in a corrupted geometry, as shown in Fig. 2 case (1) (a) and (b). Incorporating the local feature $f$ enables fast color learning and increases the representation ability of the network, and the problem is noticeably alleviated, as shown in Fig. 2 case (1), the differences between (a) and (c), (b) and (d).

**The key to reconstructing accurate geometry details.** We then introduce another case in Fig. 2 case (2). Its texture changes moderately, and the color is largely correlated with the surface normal due to diffuse reflection. Although the geometry still collapses given neither normal $n$ or feature $f$ as input in Fig. 2 case (2) (a), we can observe a reasonable reconstruction even with some geometry details in Fig. 2 case (2) (b) with only $n$ as the input. Incorporating the feature $f$ does not further reduce the Chamfer Distance (CD); instead, geometry details are missing since the learnable feature $f$ disturbs the geometry-color dependency, *i.e.*, the relationship built between the color and the surface normal, as shown in Fig. 2 case (2), the differences between (b) and (d). Considering the cons and pros of the learnable local feature, it is valuable to design a framework that leverages it for coherent shape and keeps color-geometry dependency for fine details.

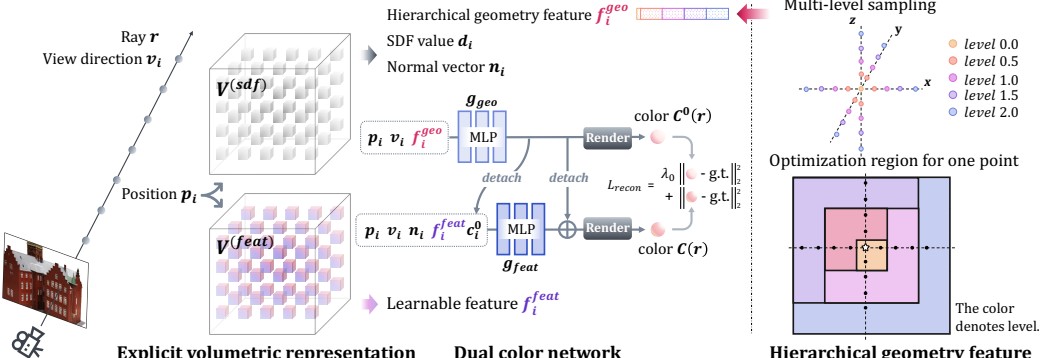

Figure 3: **Overview of key components in our model.** We adopt an explicit volumetric representation with an SDF voxel grid $V^{(sdf)}$ and a feature voxel grid $V^{(feat)}$. In the middle, we show the design for our dual color network, where $f_i^{feat}$ is the interpolated feature from $V^{(feat)}$ at point $p_i$, and $f_i^{geo}$ denotes the hierarchical feature constructed on the right. Here we show the multi-level sampling scheme and the region of grids that is affected by one point during optimization with different settings of levels.

**The reason for noisy surfaces.** For all the cases above, the results suffer from obvious noise on the surface. Compared with learning an implicit representation globally, the under-constrained voxel grids lack spatial coherence and are vulnerable to local minima, which hurts the continuity and smoothness of the surface. An intuitive idea is leveraging geometry cues from a region rather than a local point, which can be introduced in model inputs, network components, and loss functions.

## 5 METHODOLOGY

Inspired by the insight revealed in Sec. 4, we propose several key designs: 1) we adopt a two-stage training procedure that attains a coherent coarse shape (Sec. 5.1) and recovers fine details (Sec. 5.2) successively; 2) we propose a dual color network to maintain color-geometry dependency and recover precise surfaces and novel-view images; 3) we design a hierarchical geometry feature to encourage information propagation across voxels for stable optimization; 4) we also introduce smoothness priors, including a gradient smoothness loss for better visual quality (Sec. 5.3).

### 5.1 COARSE SHAPE INITIALIZATION

We initialize our SDF voxel grid $V^{(sdf)}$ with an ellipsoid-like zero level set inside a prepared region for reconstruction as in (Sun et al., 2022a). We then perform coarse shape optimization with the aid of $V^{(feat)}$ as introduced in Sec. 4. Specifically, we train a shallow MLP with both normal vector $n$ and local feature $f$ as inputs, along with the embedded position $p$ and viewing direction $v$. To encourage a stable training process and smooth surface, we propose to conduct the interpolation on a smoothed voxel grid rather than the raw data of $V^{(sdf)}$. In particular, we denote $\mathcal{G}(V, k_g, \sigma_g)$ as applying 3D convolution on the voxel grid $V$ with a Gaussian kernel, whose weight matrix follows a Gaussian distribution: $K_{i,j,k} = 1/Z \times \exp\left(-((i - \lfloor k_g/2 \rfloor)^2 + (j - \lfloor k_g/2 \rfloor)^2 + (k - \lfloor k_g/2 \rfloor)^2)/2\sigma_g^2\right)$, $i, j, k \in \{0, 1, ..., k_g - 1\}$, where $Z$ denotes a normalization term, $k_g$ denotes the kernel size, and $\sigma_g$ denotes the standard deviation. Querying a smoothed SDF value $d'$ of an arbitrary point $p$ thus becomes:

$$d' = \text{interp}(p, \mathcal{G}(V^{(sdf)}, k_g, \sigma_g)). \tag{5}$$

We use $d'$ for the ray marching integration following Eqn. 1 and Eqn. 2 and calculate the reconstruction loss. We also apply several smoothness priors as to be introduced in Sec. 5.3

### 5.2 FINE GEOMETRY OPTIMIZATION

At this stage, we aim to recover accurate geometry details based on the coarse initialization. We note that the challenges are two-fold: **1)** The study in Sec. 4 reveals a trade-off introduced by the feature voxel grid, *i.e.*, the representation capacity of the color field is improved at the sacrifice of color-geometry dependency. **2)** The optimization of the SDF voxel grid is based on trilinear interpolation

to query a 3D point. The operation brings in fast convergence, while it also limits information sharing across different locations, which may lead to local minima with degenerate solutions and a sub-optimal smoothness. We propose a *dual color network* and a *hierarchical geometry feature* to address these two issues, respectively.

**Dual color network.** The observation in Sec. 4 encourages us to design a dual color network that takes advantage of the local feature $f_i^{feat}$ interpolated from the learnable feature voxel grid $V^{(feat)}$ without losing the color-geometry dependency. As shown in Fig. 3, we train two shallow MLPs with different additional inputs besides the embedded position and view direction. The first MLP $g_{geo}$ takes the hierarchical geometry feature $f_i^{geo}$, which will be introduced later, to build the color-geometry dependency; the second one $g_{feat}$ takes both a simple geometry feature (*i.e.*, the surface normal $n_i$) and the local feature $f_i^{feat}$ as inputs to enable a faster and more precise color learning, which will in turn benefit the geometry optimization. The two networks are combined in a residual manner with detaching operations: the output of $g_{geo}$, denoted by $c_0$, is detached before input to $g_{feat}$, and the output is added back to a detached copy of $c_0$ to get the final color prediction $c$.

Outputs of both $g_{geo}$ and $g_{feat}$ are supervised by a reconstruction loss between the ground truth image and the integrated color along the ray. Specifically, the rendered colors from them are denoted as $C^0(r)$ and $C(r)$, and the overall reconstruction loss is formulated as:

$$\mathcal{L}_{recon} = \frac{1}{\mathcal{R}} \sum_{r \in \mathcal{R}} \left( ||C(r) - \hat{C}(r)||_2^2 + \lambda_0 ||C_0(r) - \hat{C}(r)||_2^2 \right), \tag{6}$$

where $\hat{C}(r)$ denotes the ground truth color, and $\lambda_0$ denotes a loss weight. $V^{(feat)}$ and the MLP $g_{feat}$ fit the scene field rapidly, while the MLP $g_{geo}$ fits the scene at a relatively slower pace. The detaching operations promote a stable optimization of $g_{geo}$ guided by the reconstruction loss of itself, which helps preserve color-geometry dependency.

**Hierarchical geometry feature.** Using the surface normal $n$ as the geometry feature for the color networks is a straightforward choice, while it takes in information only from adjacent grids of $V^{(SDF)}$. In order to enlarge the perception area and encourage information propagation across voxels, we propose to look at a larger region of the SDF field and take the corresponding SDF values and gradients as an auxiliary condition to the color networks. Specifically, for a given 3D position $p = (x, y, z)$, we take half of the voxel size $v_s$ as the step size and define its neighbours along the $X, Y, Z$ axis on both sides. Taking the $X$ axis as an example, the neighbouring coordinates are defined as $p_x^{l-} = (x^{l-}, y, z)$ and $p_x^{l+} = (x^{l+}, y, z)$, where $x^{l-} = \max(x - l * v_s, 0)$, $x^{l+} = \min(x + l * v_s, v_x^m)$, $l \in [0.5, 1.0, 1.5, ...]$ denotes the 'level' of neighbour area, and $v_x^m$ denotes the maximum of the voxel grid on $x$ axis. We then extend the definition to a hierarchical manner by concatenating the neighbours from different levels together as formulated below:

$$d_k^l = [d_k^{l-}, d_k^{l+}] = [\text{interp}(p_k^{l-}, V^{(sdf)}), \text{interp}(p_k^{l+}, V^{(sdf)})], k \in \{x, y, z\},$$
$$f_p^{sdf}(l) = [d^0, d_x^{0.5}, d_y^{0.5}, d_z^{0.5}, \cdots, d_x^l, d_y^l, d_z^l]^T, \tag{7}$$

where $d_x^l$ denotes the SDF values queried from $V^{(sdf)}$ at locations $p_x^{l-}$ and $p_x^{l+}$. When $l = 0$, $f_p^{sdf}(0) = d^0$, which is exactly the SDF value at the location $p$ itself. Then, we also incorporate the gradient information into the geometry feature. Specifically, we can obtain the gradient vector $\delta_x^l = (d_x^{l+} - d_x^{l-})/(2 * l * v_s)$. We normalize the $[\delta_x^l, \delta_y^l, \delta_z^l]$ to a L2-norm of 1, denoted as $n^l \in \mathbb{R}^3$. The hierarchical version of the normal is formulated as:

$$f_p^{normal}(l) = [n^{0.5}, \cdots, n^l]. \tag{8}$$

Finally, the hierarchical geometry feature at point $p$ for a predefined level $l \in [0.5, 1.0, 1.5, ...]$ is to combine the information above by:

$$f_p^{geo}(l) = [f_p^{sdf}(l), f_p^{normal}(l)]. \tag{9}$$

As shown in Fig. 3, $f_p^{geo}(l)$ is input to the MLP $g_{geo}$ to assist geometry learning.

## 5.3 SMOOTHNESS PRIORS

We incorporate two effective regularization terms to facilitate surface smoothness during training. **(1)** First, we adopt a total variation (TV) regularization (Rudin & Osher, 1994):

$$\mathcal{L}_{TV}(V) = \sum_{d \in [D]} \sqrt{\Delta_x^2(V, d) + \Delta_y^2(V, d) + \Delta_z^2(V, d)}, \tag{10}$$

Table 1: Quantitative evaluation on DTU dataset.

| Scan | 24 | 37 | 40 | 55 | 63 | 65 | 69 | 83 | 97 | 105 | 106 | 110 | 114 | 118 | 122 | mean |
|---|---|---|---|---|---|---|---|---|---|---|---|---|---|---|---|---|
| NeRF(Mildenhall et al., 2020) | 1.83 | 2.39 | 1.79 | 0.66 | 1.79 | 1.44 | 1.50 | 1.20 | 1.96 | 1.27 | 1.44 | 2.61 | 1.04 | 1.13 | 0.99 | 1.54 |
| IDR(Yariv et al., 2020) | 1.63 | 1.87 | 0.63 | 0.48 | 1.04 | 0.79 | 0.77 | 1.33 | 1.16 | 0.76 | 0.67 | 0.90 | 0.42 | 0.51 | 0.53 | 0.90 |
| DVGO(Sun et al., 2022a) | 1.83 | 1.74 | 1.70 | 1.53 | 1.91 | 1.91 | 1.77 | 2.60 | 2.08 | 1.79 | 1.76 | 2.12 | 1.60 | 1.80 | 1.58 | 1.85 |
| PointNeRF(Xu et al., 2022) | 0.87 | 2.06 | 1.20 | 1.01 | 1.01 | 1.39 | 0.80 | **1.04** | 0.92 | 0.74 | 0.97 | **0.76** | 0.56 | 0.90 | 1.05 | 1.02 |
| NeuS(Wang et al., 2021) | 0.83 | 0.98 | 0.56 | 0.37 | 1.13 | **0.59** | **0.60** | 1.45 | 0.95 | 0.78 | **0.52** | 1.43 | **0.36** | 0.45 | **0.45** | 0.77 |
| DVGO + NeuS | 1.24 | 0.87 | 0.74 | 0.48 | 1.20 | 1.41 | 1.113 | 1.96 | 1.44 | 0.98 | 1.13 | 1.99 | 1.62 | 0.77 | 0.62 | 1.13 |
| Ours | **0.65** | **0.74** | **0.39** | **0.35** | **0.96** | 0.64 | 0.85 | 1.58 | 1.01 | **0.68** | 0.60 | 1.11 | 0.37 | **0.45** | 0.47 | **0.72** |

Table 2: An overall comparison on surface reconstruction, novel view synthesis, and training time on DTU.

| | PSNR ↑ | SSIM ↑ | LPIPS ↓ | CD ↓ | Time (Nvidia A100) |
|---|---|---|---|---|---|
| DVGO (Sun et al., 2022a) | 31.64 | 0.916 | 0.159 | 1.85 | 4 mins |
| NeuS (Wang et al., 2021) | 29.63 | 0.892 | 0.199 | 0.77 | 5.5 hours |
| Ours | **32.16** | **0.929** | **0.144** | **0.72** | 15 mins |

where $\Delta_x^2(V, d)$ denotes the squared difference between the value of $d$th channel in voxel $v := (i; j; k)$ and the $d$th value in voxel $(i + 1; j; k)$, which can be analogously extended to $\Delta_y^2(V, d)$ and $\Delta_z^2(V, d)$. We apply the TV term above to the SDF voxel grid, denoted by $\mathcal{L}_{TV}(V^{(sdf)})$, which encourages a continuous and compact geometry.

**(2)** We also assume the surface to be smooth in a local area, and we follow the definition of the Gaussian convolution in Sec. 5.1 and introduce a smoothness regularization formulated as:

$$\mathcal{L}_{smooth}(V) = ||\mathcal{G}(V, k_g, \sigma_g) - V||_2^2, \tag{11}$$

We apply the smoothness term above to the gradient of SDF voxel grid for a *gradient smoothness loss*, denoted by $\mathcal{L}_{smooth}(\nabla V^{(sdf)})$. It encourages a smooth surface and alleviates the issue of noisy points in the free space. Notice that we can also naturally conduct post-processing on the SDF field after training, thanks to its explicit representation. For example, applying the Gaussian kernel above before extracting the geometry can further boost surface smoothness for better visualization.

Finally, the overall training loss is formulated as:

$$\mathcal{L} = \mathcal{L}_{recon} + \lambda_{tv}\mathcal{L}_{TV}(V^{(sdf)}) + \lambda_s\mathcal{L}_{smooth}(\nabla V^{(sdf)}), \tag{12}$$

where $\lambda_{tv}$ and $\lambda_s$ denote the weights for the corresponding loss terms.

# 6 EXPERIMENTS

**Experimental setup.** We use the DTU (Jensen et al., 2014) dataset for quantitative and qualitative comparisons and show qualitative results on several challenging scenes from the BlendedMVS (Yao et al., 2020) dataset. We include several baselines for comparisons: 1) IDR (Yariv et al., 2020), 2) NeuS (Wang et al., 2021), 3) NeRF (Mildenhall et al., 2020), 4) DVGO (Sun et al., 2022a), 5) Point-NeRF (Xu et al., 2022). The results of 1), 2), and 3) are taken from the original papers (Yariv et al., 2020; Wang et al., 2021); for 5), we use the neural point for evaluation. We provide a clean background for all the methods for a fair comparison. Experimental results with non-empty backgrounds and comparisons with more methods (Schönberger et al., 2016; Oechsle et al., 2021; Yariv et al., 2021) are included in the supplementary materials. Please also refer to the supplementary materials for further descriptions of the datasets, baseline methods, and implementation details.

## 6.1 COMPARISONS

The quantitative results for surface reconstruction on DTU are reported in Table 1. Quantitative experimental results show that we achieve lower Chamfer Distances than previous methods under the same setting. We conduct qualitative comparisons on both DTU and BlendedMVS in Fig. 4 and Fig. 5, respectively. DVGO shows poor reconstruction quality with noise and holes since it is designed for novel view synthesis rather than surface reconstruction. NeuS and ours show accurate and continuous surface recovery in a variety of cases. In comparison, NeuS, as a fully implicit model, naturally benefits from the intrinsic continuity and encourages smoothness in local areas,

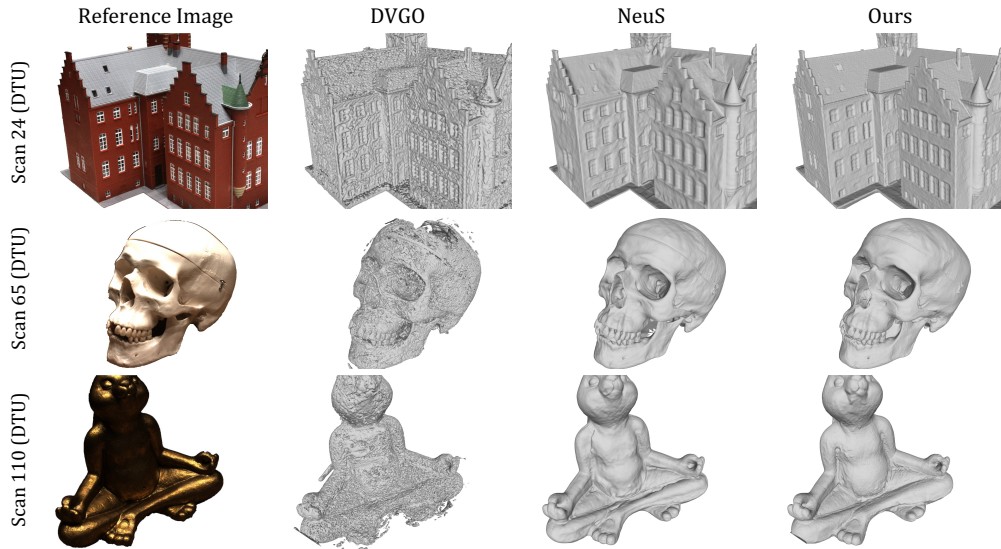

Figure 4: Qualitative comparisons on the DTU dataset. See more scenes in supplementary materials.

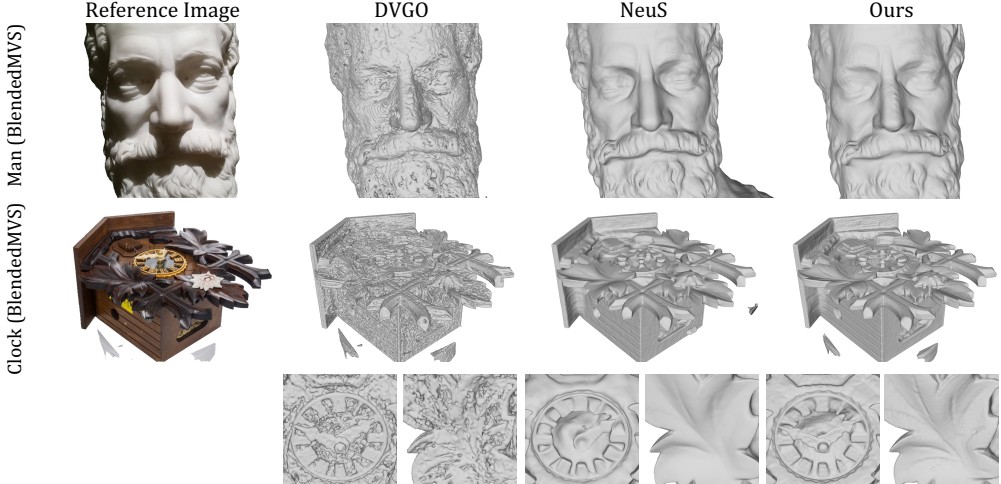

Figure 5: Qualitative comparisons on the BlendedMVS dataset. See more scenes in supplementary materials.

while it sometimes fails to recover very thin geometry details due to over-smoothing. In contrast, our method is superior in recovering fine geometry details thanks to our designs in Sec. 5.

We further perform a more extensive evaluation of our method on surface reconstruction, novel view synthesis, and training time in Table 2. Our method outperforms DVGO and NeuS on both surface reconstruction and novel view synthesis by a clear margin on all the metrics. Notably, our method achieves around **20x** speedup compared to NeuS for producing high-quality surface reconstruction.

## 6.2 ANALYSIS

In this section, we carry out a series of ablation studies to evaluate each technical component.

**The effect of the dual color network and a hierarchical geometry feature.** As shown in Table 3, both techniques individually work well on the baseline model, and a combination of them produces the best result. **1)** The effect of dual color network can be directly sensed in the improvement of image rendering quality, as can be seen from the comparison of roof textures in Fig. 6. An accurate color field and the color-geometry dependency will promote geometry learning, as can be observed from the comparison of roof geometries (viewed in normal images) in Fig. 6. Experimental results in Table 4 also validate the effectiveness of the design introduced in Sec. 5.2, including the residual color and detachment. **2)** Hierarchical geometry feature directly promotes an accurate

GT (test view)   w/o dual color network   w/o geometry feature   Full model

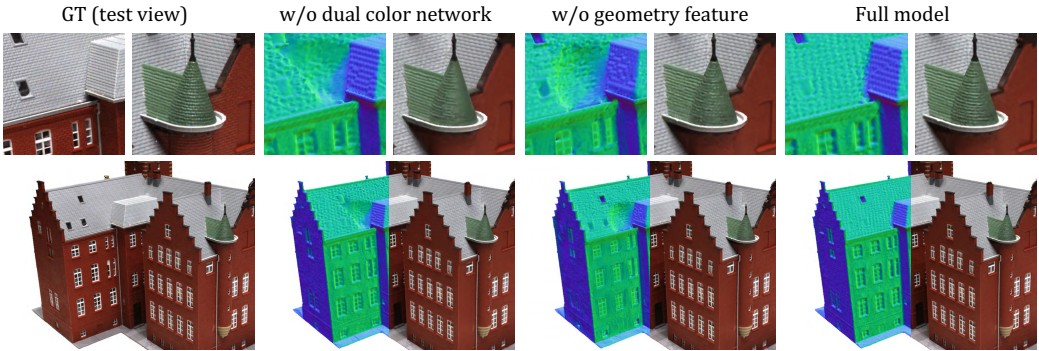

Figure 6: The dual color network learns the color field for complex scenes well and preserves color-geometry dependency, which facilitates geometry learning (see the roofs); the hierarchical geometry feature promotes accurate surface reconstruction (see the windows).

Table 3: Ablation over the effect of dual color network and hierarchical geometry feature.

| CD | 0.91 | 0.79 | 0.77 | **0.72** |
|---|---|---|---|---|
| Dual | | ✓ | | ✓ |
| Hierarchical | | | ✓ | ✓ |

Table 4: Ablation over the *Residual* and *Detach* designs of the dual color network (Sec. 5.2).

| CD | 0.77 | 0.75 | 0.75 | **0.72** |
|---|---|---|---|---|
| Residual | | ✓ | | ✓ |
| Detach | | | ✓ | ✓ |

surface reconstruction, as demonstrated by results in Table 3 and the difference between normal images of Fig. 6. We also explore different design details, including the level selection and the effects of gradient and SDF value in supplementary materials.

**Ablation over smoothness priors.** We make efforts to encourage the continuity and smoothness of the reconstructed surface at different stages. As shown in Fig. 7 (a), during the coarse shape initialization stage, the naive solution produces holes and noises. Applying the Gaussian convolution substantially alleviates the problem and leads to a more compact geometry. Regularization terms including the TV and our gradient smoothness loss would further encourage a clean and smooth surface to provide a good initialization for the next stage. Fig. 7 (b) shows that during the fine geometry optimization stage, the regularization terms also help maintain surface smoothness. Finally, as shown in Fig. 7 (c), post processing on a trained model can promote surface smoothness for a better visualization quality and maintain an accurate structure at the same time. An ablation study on the effects of the SDF TV term and our gradient smoothness loss is in the supplementary materials.

(a) Training stage-1            (b) Training stage-2            (c) Inference stage

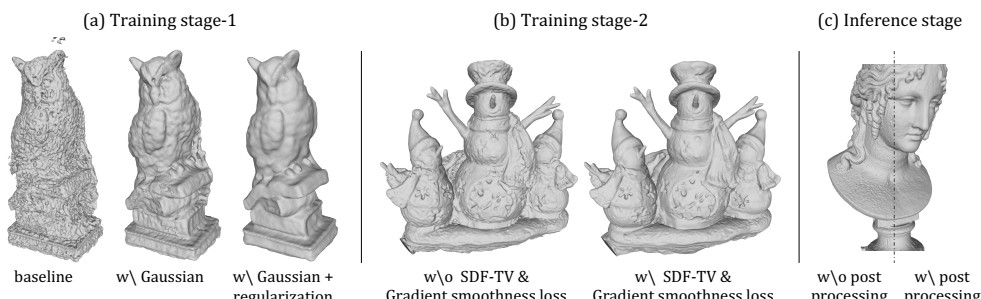

baseline   w\ Gaussian   w\ Gaussian +          w\o SDF-TV &        w\ SDF-TV &        w\o post   w\ post
                          regularization   Gradient smoothness loss   Gradient smoothness loss   processing   processing

Figure 7: Studies on technical components that encourage surface smoothness during the (a) coarse shape initialization, (b) fine geometry optimization, and (c) post-processing stage.

## 7 CONCLUSION

This paper proposes **Voxurf**, a voxel-based approach for efficient and accurate neural surface reconstruction. It includes several key designs: the two-stage framework attains a coherent coarse shape and recovers fine details successively; the dual color network helps maintain color-geometry dependency, and the hierarchical geometry feature encourages information propagation across voxels; effective smoothness priors including a gradient smoothness loss further improve the visual quality. Extensive experiments show that Voxurf achieves high efficiency and high quality at the same time.

## 8 ACKNOWLEDGEMENT

This work is supported by Shanghai AI Laboratory, NTU NAP, MOE AcRF Tier 2 (MOE-T2EP20221-0012), and under the RIE2020 Industry Alignment Fund – Industry Collaboration Projects (IAF-ICP) Funding Initiative.

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
