# OpenReview forum: "Voxurf: Voxel-based Efficient and Accurate Neural Surface Reconstruction"
_ICLR.cc/2023/Conference — ICLR 2023 notable top 25%_

### Official Review · Reviewer_kZqD · 2022-10-23

**Confidence:** 4
**Correctness:** 3
**Technical Novelty And Significance:** 2
**Empirical Novelty And Significance:** 3
**Recommendation:** 6

**Clarity, Quality, Novelty And Reproducibility:**

The paper is clearly written and easy to follow. The work is of high quality and contains original ideas, but I have concerns, listed above, on whether it is significant, or an incremental step compared to recent literature.

Some important information, including the choice of the dimensions of the voxel grid and the way to obtain an explicit representation of the surfaces (for the geometric criteria used in the quantitative results), can only be found in the supplement.

Related work is presented briefly but clearly. There are some important omissions, including the following:
Lingjie Liu, Jiatao Gu, Kyaw Zaw Lin, Tat-Seng Chua, and Christian Theobalt. Neural sparse voxel fields. In NeurIPS,2020.
Vincent Sitzmann, Justus Thies, Felix Heide, Matthias Nießner, Gordon Wetzstein, and Michael Zollhoefer. Deepvoxels: Learning persistent 3d feature embeddings. In CVPR, 2019.
The unrefereed (arXiv) versions of several papers are cited instead of the refereed ones. For example, the DVGO paper by Sun et al. has been published at CVPR 2022. The same is true for the work of Darmon et al. and PointNeRF by Xu et al. The paper by Lombardi et al. has been published in the ACM Transactions on Graphics. All references for which this happens should be corrected.

The term “feature” is overloaded in Section 5, in particular in Section 5.2. It is used both for the “geometric feature” f^geo and the “local feature” f^feat.

There are some typos scattered throughout the paper. The following are a few examples.
p. 2: “under-constraint” should be “under-constrained”
p. 6, below (7): “gain” seems out of place. “Obtain” would be better. The following line includes “the approximate normal vector n^l in R^3 is to normalize” which can be improved.
p. 6, first line of Sec. 5.3: I suggest specifying that there are two, not several, regularization terms. Several suggests that a lot of heuristics are needed for convergence.


**Strength And Weaknesses:**

Strengths

The paper presents an approach for neural surface reconstruction employing an SDF representation decomposed in voxels. To accelerate convergence, the authors adopt a two-stage training procedure. I consider this a strength of the paper, but not a major one. It is not surprising that such a scheme is fast and also effective, as long as the coarse grid is of sufficient resolution.

The dual color network is novel to the best of my knowledge.  It comprises two branches that aim to synthesize accurate colors in novel viewpoints. The design is motivated by the observation that color and geometry drift apart as larger contexts are considered for geometric optimization. The two branches are based on different feature inputs, but also share information, to synthesize correct views while contributing to geometry optimization and fast convergence.

The hierarchical geometry feature enables information propagation across voxels by considering neighborhoods of different size, mitigating a limitation of voxel-based formulations. In my opinion, this is the most important contribution of the paper.

The smoothness priors are shown to be effective experimentally. This indicates that the dual color network and the hierarchical geometric features do not oversmooth. It may have been possible to achieve sufficient regularization with the other components of the network, but since this is not the case, the additional terms defined in Section 5.3 contribute. (The impact of all network components and their interactions on convergence time is unclear.)

The proposed approach is faster than many alternatives relying on global representations, but see also below for a relevant weakness.

An important strength of the paper is that ablations studies for all aspects are presented in the main paper and the supplement.

Weaknesses

A minor question, not necessarily a weakness, in my opinion is the loss function associated with the dual color network. It is unclear to me why the loss aims at having both branches achieve perfect view synthesis independently. It may be more advantageous to require that the combination of the two branches achieves this goal.

The method is fast, but it is much slower than the naïve baseline constructed by the authors at the end of Section 3, specifically DVGO with an SDF representation. This is not surprising due to the second stage of training, but weakens the argument that the proposed method somehow advances the efficiency of neural surface reconstruction methods. It seems to be on the efficiency-fidelity curve of voxel-based approaches.

An important weakness in the evaluation is that the comparison with DVGO, which is similar to the naïve baseline, is performed on different datasets than those used by the authors of DVGO. In particular, four of the five datasets used in the DVGO paper are missing, with only BlendedMVS used in both papers. The four missing datasests are: Synthetic-NeRF (Mildenhall et al., ECCV 2020), Synthetic-NSVF (Liu et al., NeurIPS 2020 – see below), Tanks & Temples (Knapitsch et al., SIGGRAPH 2017) and DeepVoxels (Sitzmann et al., CVPR 2019). Some of these papers are not even cited. (See below.) PointNeRF by Xu et al., which is cited, presents results on the DTU dataset with better PSNR, SSIM and LPIPS than the proposed method, while also being faster.

The surface reconstruction results on DTU, as measured by the average of the Chamfer distances between the reconstructed and ground truth points in both directions, are much weaker than those of recent MVS methods. MVS reconstructions are in the order of 0.3-0.35 mm while the proposed method is at 0.72 mm. (MVS papers use the same test set on DTU, which may be different than the one used here and by DVGO, but I assume that the statistics should be similar on average.) Chamfer distance may not measure interesting properties of the surfaces such as smoothness and watertightness, but the authors accept it as a metric.

Based on the above, novel view synthesis should be the focus of the evaluation, since NeRF-style methods are not ready to be used for explicit surface reconstruction yet. Therefore, additional experiments on datasets widely used in the NeRF literature are needed to demonstrate the advantages of the proposed method.


**Summary Of The Paper:**

The paper presents a fast, voxelized algorithm for implicit surface reconstruction and novel view synthesis. It addresses limitations of previous methods in preserving fine geometric details by a two-stage training procedure starting with a coarse resolution grid and moving on to finer resolution, a two-branch color network, and a hierarchical geometry feature that promotes inter-voxel coherence. Surface reconstruction and novel view synthesis results are presented on the DTU and BlendedMVS datasets, along with comparisons to relevant methods. Extensive ablation studies are also presented.

**Summary Of The Review:**

At least two of the key ideas of the paper, specifically the dual color network and the hierarchical feature design, are novel, but it is unclear to me how effective they are beyond the context of the algorithm presented here. I was hoping that a thorough evaluation, including comparisons with previous work, would answer this question. The results presented in the paper are extensive, but they fall short of this because they are not performed on the same datasets as the most relevant previous methods. Important relevant work, such as PointNeRF, is omitted from the evaluation. I consider this kind of evaluation necessary for making a decision in favor of acceptance.

+++ UPDATE DURING DISCUSSION +++

I have been convinced by the authors’ responses and I have upgraded my rating of the paper.

---

> ### Author Response · Authors · 2022-11-18
> **Response to Reviewer kZqD (Part 1).**
>
> **Question 1: It may be more advantageous to require that the combination of the two branches achieves this goal.**
>
> **Answer 1:** Thank you for the question. The output of the second branch is already combined results, and we require it to achieve a perfect view synthesis. The first loss term is designed to facilitate color-geometry dependency since it directly links the color with hierarchical geometry features without being disturbed by the learnable feature grid.
>
> **Question 2: The method seems to be on the efficiency-fidelity curve of voxel-based approaches.**
>
> **Answer 2:** Thank you for the inspiring feedback. For our method itself, there indeed exists a trade-off between efficiency and fidelity. We explore the trade-off by changing the voxel grid resolution as in Table R6. When we adopt the resolution of 192^3 in Voxurf, the training time becomes 12min with a CD of 0.75. It is much better than the DVGO+SDF (DVGO + NeuS)  baseline, which takes 12min but only achieves a CD of 1.13.
>
> **Question 3: Novel view synthesis should be the focus of the evaluation since NeRF-style methods are not ready to be used for explicit surface reconstruction yet.**
>
> **Answer 3:** Thank you for your detailed discussion and suggestion. However, we would like to explain the motivation and advantages of NeRF-style methods in surface reconstruction (Oechsle et al., 2021; Yariv et al., 2021; Wang et al., 2021; Darmon et al., 2021). We also highlight that the main idea and core contributions of this paper focus on surface reconstruction, as stated in our title.
>
> First, MVS approaches such as the widely-used COLMAP  (Schonberger et al., 2016, ) usually rely on complex pipelines, e.g., depth estimation/optimization, depth fusing, and surface reconstruction from point clouds (Kazhdan et al., 2006). Errors may accumulate at each stage, and the output mesh is usually incomplete, especially for non-Lambertian surfaces as they can not handle view-dependent colors (Yariv et al., 2021). In contrast, NeRF-style methods overcome the problems above by directly producing a smooth, watertight, and accurate zero-level set. When we extract a watertight mesh from COLMAP, the results are worse than NeRF-style models (see Table R2).
>
> Second, learning-based MVS methods (Yao et al., 2018; Gu et al., 2020; Wang et al., 2021) have achieved impressive results on DTU recently, while they are mostly trained on the specific dataset supervised by ground truth depth maps. This introduces extra time consumption and hinders their generalization to custom data in different downstream applications. Unsupervised approaches (Darmon et al., 2021) are less competitive in performance. In comparison, NeRF-based methods do not require any data or ground-truth depth for pre-training, and they can be flexibly applied to custom data in any domain and produce high-quality reconstruction results.
>
> In summary, recent approaches of NeRF-style surface reconstruction are valuable and worth exploring. This paper follows the benchmark and focuses on accurate and efficient surface reconstruction.
>
> J. L. Schönberger, E. Zheng, M. Pollefeys, and J.-M. Frahm. Pixelwise view selection for unstructured multi-view stereo. In European Conference on Computer Vision, 2016.
>
> M. Kazhdan, M. Bolitho, and H. Hoppe. Poisson Surface Reconstruction. In A. Sheffer and K. Polthier, editors, Symposium on Geometry Processing. The Eurographics Association, 2006.
>
> Yao Yao, Zixin Luo, Shiwei Li, Tian Fang, and Long Quan. MVSNet: Depth inference for unstructured multiview stereo. In European Conference on Computer Vision, 2018.
>
> Xiaodong Gu, Zhiwen Fan, Siyu Zhu, Zuozhuo Dai, Feitong Tan, and Ping Tan. Cascade cost volume for high-resolution multi-view stereo and stereo matching. In Conference on Computer Vision and Pattern Recognition, 2020.
>
> Fangjinhua Wang, Silvano Galliani, Christoph Vogel, Pablo Speciale, and Marc Pollefeys. Patchmatchnet: Learned multi-view patchmatch stereo. In Conference on Computer Vision and Pattern Recognition, 2021.
>
> Franc¸ois Darmon, Ben´ edicte Bascle, Jean-Cl ´ ement Devaux, ´ Pascal Monasse, and Mathieu Aubry. Deep multi-view stereo gone wild. In Int. Conf. on 3D Vision, 2021.
>
> *(Answers to the other questions are presented in part 2.)*

---

> > ### Author Response · Authors · 2022-11-18
> > **Response to Reviewer kZqD (Part 2)**
> >
> > **Question 4:  (If the evaluation focuses on novel view synthesis based on Question 3,) several datasets used in DVGO are missing, and another acceleration method for novel view synthesis should be included for comparison.**
> >
> > **Answer 4:** Thank you for the valuable suggestions. We explain in Answer 3 that we mainly focus on surface reconstruction in this paper, and we adopt the standard evaluation database following previous approaches such as NeuS. The reason why we didn’t involve the other four datasets (Mildenhall et al., 2020; Liu et al., 2020; Knapitsch et al., 2017; Sitzmann et al., 2019) from DVGO is that three of them do NOT have publicly available 3D ground truth to evaluate surface reconstruction. Nevertheless, we follow your suggestions to add experiments on the additional datasets, and the results are presented in Sec. E, a new section in the supplementary material, which also confirms the advantages of our method.
> >
> > We also involve Point-NeRF (Xu et al.) for comparison as suggested by the reviewer, which is an impressive work on NeRF acceleration based on an innovative neural point representation. This approach is proposed for the NVS task, and we manage to obtain the neural points after the finetuning stage and use Poisson surface reconstruction (Kazhdan et al., 2006) to extract a surface from the points. We add a new section in the supplementary material (Sec. F) to analyze the results.
> >
> > We will try to adopt PointNeRF in the benchmark and perform a comprehensive evaluation on both surface reconstruction and rendering in the final version.
> >
> > **Question 5:  Some suggestions on writing and typos.**
> >
> > **Answer 5:** Thank you for the thoughtful suggestions. We have corrected typos and citations. We will carefully revise overloaded words in the final version.

---

### Official Review · Reviewer_8Ao6 · 2022-10-24

**Confidence:** 3
**Correctness:** 3
**Technical Novelty And Significance:** 3
**Empirical Novelty And Significance:** 3
**Recommendation:** 8

**Clarity, Quality, Novelty And Reproducibility:**

Clarity:
- The paper is mostly easy to follow.
- I did find it a bit confusing that the motivation in Section 4 is presented separately from the solution in Section 5. Might be worth at least hinting on what is coming later: e.g. does adding features actually help and if not what would help with appearance-geometry consistency. Explanations on dual color network are not very formal, and it is not fully clear if there are any guarantees on why that would work? E.g. it is possible that the model simply ignores some of the inputs?

Novelty:
- The method is indeed very similar to DVGO, even though it employs a different (also already existing) volume rendering scheme. It is not clear if the paper contains sufficient additional insights: the dual color network and geometry features sort of make sense, but (arguably) seem like ad-hoc solutions without formal reasoning behind those.

Quality/Misc:
- The paper does not seem to have significant issues with quality.
- The claimed speed up is a bit confusing, considering that DVGO runs much faster than that? Yes, it is much worse in CD - but would that still be the case for DVGO+little bit of smoothness priors?


**Strength And Weaknesses:**

Strength:
+ The overall approach is reasonable - it makes sense that one needs more capacity and some notion of locality to optimize
+ Quantitative results are pretty convincing: the method seems to be outperforming the strong baselines methods it builds upon - NeuS and DVGO.
+ Authors claim 20x improvement over fully implicit representation - which is not surprising (also see quality/misc below) but still impressive.

Weaknesses:
- The method does "just" combine NeuS objective with explicit voxel-based representation from DVGO, plus some additional (primarily smoothness and consistency promoting) inductive biases and priors in architecture and optimization process.
- The effects of smoothness priors are not thoroughly evaluated - in particular it would be important to know if using naive combination of DVGO+NeuS and smoothness already leads to good results.
- (minor) Qualitative results on NVS are quite similar to DVGO (it is worth looking at the supplementary material). In fact, I find on some of the examples DVGO has better specularities (see Scan 114) .
- (minor) Similarly, geometry results seem to be very similar to NeuS, in some cases (it seems) NeuS providing less noisy reconstructions: snowman's nose on Scan 69, can on Scan 97.


**Summary Of The Paper:**

This paper introduces voxurf - a method for surface reconstruction that combines explicit voxel representations with SDF-based volume rendering. On top of naive combination of existing techniques, the key ideas are: employing a coarse-to-fine strategy, such that coarse geometry is estimated first; a dual color network, which is supposed to encourage better consistency between appearance and geometry, adding "hierarchical features" and smoothness priors, which are meant to improve propagation of information across different voxels and make optimization easier. Quantitative results are convincing, and method seems to provide a significant speed up over (some) competitive.


**Summary Of The Review:**

This work presents a well-performing and efficient method for neural surface reconstruction, but it is not fully clear if it contains enough non-trivial ideas. I still believe that the overall method is interesting, and could be valuable to the community. Thus I am leaning more towards accept.
---
After going through author's rebuttal, and looking at the additional ablation studies they pointed out, I am recommending accepting this work to ICLR.

---

> ### Author Response · Authors · 2022-11-18
> **Response to Reviewer 8Ao6.**
>
> **Question 1: The effects of smoothness priors are not thoroughly evaluated. The method does "just" combine NeuS with DVGO.**
>
> **Answer 1:** Thank you for your constructive suggestion. We conduct experiments to study the effect of smoothness constraints on the SDF + DVGO baseline, and we add a new section (Sec. C) in the supplementary material for discussion. Starting from the baseline model, we study the effect of the smoothness priors and their combinations proposed in the paper. Both the quantitative results and the qualitative results confirm that the smoothness constraints alone are not sufficient to produce accurate surface reconstruction, as apparent artifacts still exist (Fig. S3), and the CD is obviously higher than our method (Tab. R7). Meanwhile, adding strong smoothness constraints produces over-smoothed results, and it also substantially increases the time consumption of the baseline. Our method does not need to perform the heaviest smoothness constraints (e.g., Gaussian kernel for training) throughout the training due to the other key designs. More experimental results and analysis can be found in Sec. C.
>
> In summary, the smoothness constraints are NOT the only key point for good results. Furthermore, we have included a series of ablation studies (Sec. 6.2 in the main text and Sec. B.3 in the supplementary materials), where we carefully examine the effect of each component and their design details. We prove that all these effective modules contribute to the final performance of our method, which is not trivial.
>
> **Question 2: Qualitative results on NVS are quite similar to DVGO.**
>
> **Answer 2:** Thank you for the careful observations and comparisons.
>
> First, for surface reconstruction approaches, achieving a high NVS performance is a more challenging goal than methods that merely focus on NVS, like NeRF and DVGO. For example, NeuS is lower in PSNR than NeRF (Table R1 in the appendix), and VolSDF (Yariv et al., 2021) is lower in PSNR than NeRF++ (Table 2 in their paper). It is because NVS is easier with a less constrained geometry due to the ambiguity in volume rendering and the view-dependent color.
>
> Besides, the main focus of this work is not to further improve NVS of DVGO, but to perform surface reconstruction that is both *efficient* and *accurate*, which none of the previous works can achieve.
>
> **Question 3: Geometry results seem to be similar to NeuS, and in some cases (it seems) NeuS provides less noisy reconstructions.**
>
> **Answer 3:** Thank you for your careful observations and comparisons. A variance does exist when evaluating all the cases in the dataset, while on average, our method outperforms NeuS by a 6.5% reduction of CD error. More importantly, we achieve this performance with 20x speed up than NeuS, which is the main contribution of the paper. NeuS is based on a fully implicit representation, which naturally produces smooth results with less noise while removing fine geometry details at the same time.
>
> **Question 4: Some issues with clarity.**
>
> **Answer 4:**  Thank you for the valuable feedback and sorry for the confusion. We will carefully design the transition between Sec. 4 and Sec. 5 for better clarity. For the dual color network, the first branch $g_{gro}$ is mainly responsible for building the color-geometry dependency since it directly connects the hierarchical geometry feature (constructed from the SDF grid) with the color loss, and the optimization of the SDF grid will not be disturbed by the learnable feature grid; The second branch $g_{feat}$ takes the local feature as inputs to enable more precise color learning, which in turn benefits geometry optimization. When we turn off the supervision for the first branch, we see an obvious loss of geometry details (e.g., the eyes and feathers of an owl sculpture); when we turn off the supervision for the second branch, some geometrical flaws will appear in places with complex lighting and specular color. We will add more detailed explanations in the final version.

---

### Official Review · Reviewer_7aTz · 2022-10-25

**Confidence:** 3
**Correctness:** 3
**Technical Novelty And Significance:** 3
**Empirical Novelty And Significance:** 3
**Recommendation:** 8

**Clarity, Quality, Novelty And Reproducibility:**

**Quality**

This paper demonstrates a high-quality research process in which design decisions are well-motivated and experiments are devised to clearly test the impact of the design. Comparisons with prior work are reasonably exhaustive, and effective analysis is performed on the experiments and ablation studies.

**Clarity**

The paper is organized and reasonably easy to follow. The paper regularly draws links between motivations, designs, and analyses, which helps maintain a linear relationship between insight and outcome.

**Originality**

This work builds heavily on prior works such as DVGO (Sun et al., 2021) and NeuS (Wang et al., 2021). It also applies some well established components, like a Total Variation regularizer (Rudin & Osher, 1994). However, it incorporates its own new ideas, such as the hierarchical geometric feature to mitigate the spatial incoherence of volumetric representations and low pass filtering on the SDF voxel grid prior to interpolation. The additional insight provided by the architecture design study lends credence to the originality of the proposed solution.

**Strength And Weaknesses:**

**Strengths**

* The choice of baseline as the straightforward combination of DVGO (Sun et al., 2021) and NeuS (Wang et al., 2021) is effective for demonstrating the performance improvements that Voxurf offers. The improved speed and reconstruction accuracy results helps to answer the gut-response question of "why not just combine them?".

* The architecture design analysis in Section 4 and references to it in the methodology explanation in Section 5 is very helpful to motivate and explain design decisions. This paper takes great pains to justify the network architecture and loss function. In many papers, this insight is often omitted or overly brief, which limits the ability of future researchers to learn from the work.

* The proposed idea to low-pass filter the SDF volume with a Gaussian kernel, in particular, seems to have an important effect on producing smooth surface reconstructions. Although, it may be fair to question whether the emphasis on smooth outputs is overdone, as Figure 7 appears to show fine detail being smoothed away when combining both TV regularization and the Gaussian filtering. Evaluating the impact of each of the two smoothing components is an ablation study that is missing, and the paper would benefit from including.


**Weaknesses**

* Section 4 describes the key to maintaining a coherence coarse shape to be the local feature, pointing out the poor geometry in Figure 2 case (1)(b) as an example of how ignoring the local feature leads to noisy reconstructions. On the other hand, Figure 2 case (2)(b) uses the same setup but provides top performance (and also qualitatively looks pretty clean). Moreover, it strongly preserves the coarse shape and maintains fine detail. This figure seems to undercut the point of the claim.

* Table 2 presents a comparison of surface reconstruction and novel view synthesis on the DTU dataset. Although Voxurf is proposed to combine the speed benefits of DVGO with the output quality of NeuS, it is an interesting result to see that DVGO outperforms NeuS on novel view synthesis metrics at baseline. Moreover, Voxurf is only marginally better than DVGO for novel view synthesis despite taking 3.5x as long to train. Now, in terms of surface reconstruction metrics, Voxurf supports its claims by outperforming NeuS significantly in reconstruction quality in a fraction of the time. However, this discrepancy ought to be explained. Is it a problem with the expressiveness of the PSNR and SSIM metrics for NVS? Perhaps there is diminishing returns in NVS as reconstruction accuracy increases?

* Both DTU and BlendedMVS consist of fairly small scenes. One of the stated benefits of volumetric representations is the faster convergence speed, especially for large scenes with many cameras. It would improve the case for the proposed method to evaluate it on larger scene datasets such as ScanNet (Dai et al., 2017). That would provide stronger support to differentiate it from methods like Instant-NGP (Muller et al., 2021) or NerfFusion (Zhang et al., 2022).

**Summary Of The Paper:**

This paper presents a hybrid voxel-grid- and MLP-based neural surface reconstruction method called Voxurf, that is both fast and accurate. It uses the volumetric representation of a voxel grid to expedite training, while proposing a two-stage training process, a novel network that captures the relationship between color and geometry, and a more expressive SDF feature, all in order to improve reconstruction accuracy.

The paper also provides a study on architecture design for hybrid volume/MLP geometry estimation models. This study motivates the insight behind the design decisions in the proposed method, and also provides clarity on how this work differentiates itself from prior research, such as DVGO (Sun et al., 2021) and NeuS (Wang et al., 2021).

Experiments are performed on two common, relevant datasets, DTU and BlendedMVS, and the results demonstrate improvements in surface reconstruction quality with faster convergence times. Furthermore, ablation studies are included to test individual aspects of the proposed architecture.

**Summary Of The Review:**

This paper effectively makes the case for Voxurf as an efficient solution for high quality neural surface reconstruction. The paper is well motivated, sufficiently evaluated, and clear with its insights. There are areas the paper could be improved, specifically in building out some of the experimental analysis further. Nevertheless, I would recommend this paper for acceptance to ICLR.

---

> ### Author Response · Authors · 2022-11-18
> **Response to Reviewer 7aTz.**
>
> **Question 1: About the explanations of Figure 2 and the key points in Section 4.**
>
> **Answer 1:** Thank you for the valuable feedback, and sorry for the confusion. Case (1) and Case (2) are two representative while different examples. The different behaviors between them reveal both the advantages and disadvantages of the local feature $f$ and surface normal $n$.
>
> Specifically, we describe in the paper that Case (1) represents “a complex scene with different materials, high-frequency textures, and view-dependent lighting information”. The local feature can increase the representation ability of the network and help maintain a coherent coarse shape. In contrast, in case (2), the texture changes moderately, and the color is largely correlated with the surface normal. Thus, the geometry-color dependency introduced by surface normal input is beneficial in this case. Unlike Case (1), the local feature can disturb the geometry-color dependency in Case (2) and diminish geometry details.
> To this end, how to deal with the potential conflict and take advantage of different designs to reconstruct both a coherent coarse shape and fine details becomes a challenging problem, which inspires us to propose unique designs in the paper.
>
>
> **Question 2: The discrepancy between novel view synthesis and surface reconstruction.**
>
> **Answer 2:** This is an interesting observation. We believe that the NVS and surface reconstruction tasks are correlated but can still perform differently. One observation is that NVS can perform very well despite bad surface reconstruction, as in DVGO and NeRF. This is caused by the geometry-color ambiguity of NeRF, i.e., small variation of geometry can lead to the same rendered color due to the ambiguity of volume rendering and the view-dependence design of NeRF. But it changes the surface reconstruction results. Thus, the significant improvement in the surface reconstruction of Voxurf compared to DVGO does not necessarily lead to a boost in NVS. Some additional objectives (e.g., the smoothness priors) also affect the optimization of the color loss. However, the main focus of this work is not to further improve NVS of DVGO. This work aims at performing surface reconstruction for both *efficient* and *accurate*, which none of the previous works can achieve.
>
> **Question 3: It would improve the case for the proposed method to evaluate it on larger scene datasets such as ScanNet (Dai et al., 2017).**
>
> **Answer 3:** Thank you for the valuable and constructive suggestion.
>
> Reconstruing a large-scale scene is quite different from the object-centric scene. Popular object-centric reconstruction approaches, e.g., VolSDF (Yariv et al., 2021), NeuS (Wang et al., 2021), suffer from a significant performance drop for larger and more complex scenes, in particular in less-observed and textureless areas. Recent works, e.g., ManhattanSDF (Guo et al., 2022), MonoSDF (Yu et al., 2022) introduces extra geometry constraints and specific designs for large-scale scene reconstruction. As a counterpart method for NeuS, it is non-trivial to adopt our Voxurf on ScanNet (Dai et al., 2017) dataset. We are trying our best to apply Voxurf to ScanNet, which will still need more time for high-quality reconstruction results. We will carefully study the problem and add a section to discuss the application of our method to larger scene datasets like ScanNet in the final version.
>
> Yu Z, Peng S, Niemeyer M, et al. MonoSDF: Exploring Monocular Geometric Cues for Neural Implicit Surface Reconstruction. arXiv preprint arXiv:2206.00665, 2022.
>
> Guo H, Peng S, Lin H, et al. Neural 3D Scene Reconstruction with the Manhattan-world Assumption, In Conference on Computer Vision and Pattern Recognition. 2022.
>
>
> **Question 4: Ablation study over smooth priors, i.e., TV regularization and the Gaussian filtering.**
>
> **Answer 4:** Thanks for the thoughtful suggestion. We conduct the quantitative comparisons in the supplementary materials (Sec. B.3: Ablation over smoothness priors). The experimental results show the effectiveness of both TV regularization and the Gaussian filtering for lower chamfer distance error. What's worth noting is that the third figure in Fig. 7 shows the results by the end of the *coarse stage*, where our goal is to have a coherent coarse shape; we do not involve Gaussian kernel for training in the fine stage, and it will not do harm to the geometry details.

---

### Official Review · Reviewer_VHoF · 2022-10-25

**Confidence:** 4
**Correctness:** 3
**Technical Novelty And Significance:** 2
**Empirical Novelty And Significance:** 2
**Recommendation:** 6

**Clarity, Quality, Novelty And Reproducibility:**

-- The overall idea is clear, although it might not be easy to understand some part in the framework. I have listed some issues that need to be clarified.
-- It might not be easy to implement the method by just looking into the paper, but the authors have claimed to release the code.
-- The method is novel, not but very significant contributions in proposing very exciting new insights.

**Strength And Weaknesses:**

Strength:
-- First of all, this paper has addressed a very critical issue when training Nerf, which is the extremely high computational cost. They have adopted the Direct Voxel representation and the key insight is to design a dual color network to model the color-geometry dependency. They have achieved better performance on fast convergence as compared with NeuS, and more accurate geometry as compared with DVGO and the naive combination of DVGO and NeuS.


Weakness:
-- I have some concerns about the assumption on color-geometry dependency. In some cases or scenes, the surface texture is not correlated with the geometry/normal. For example, we could simply have a plane with printed textures. I wonder how could this kind of situation being handled by the proposed framework.
-- After checking all the implementation details, I still didn't get the idea of how to render images just from geometry branch. It is rather unclear for this part.
-- The smoothness term is a bit tricky. How to tune the weighting parameters to stop over-smoothing?
-- A straightfoward implementation is to directly replace the density prediction with SDF in DVGO paper. And the authors have demonstrated this baseline in Fig.1. But I want to know what if we apply this smoothness constraints in this baseline method, will we also get very good geometry as well as fast convergence. For my understanding, it seems that the smoothness constraints have played an important role to have good reconstruction on the geometry.
-- It would be better to also discuss limitations of the proposed method.

**Summary Of The Paper:**

This paper deals with the problem of fast and accurate surface reconstruction by combining the explicit volumetric based representation with the implicit function. They further improve acurracy of surface geometry reconstruction by 1) enforcing color-geometry dependency with a dual color network, 2) including hierachical geometry feature, 3) incorporating smoothness constraints in the SDF field. They have demonstrated superior performance on the reconstructed geometry and rendered images, and achieved relatively fast convergence as compared with baseline method on Nerf.

**Summary Of The Review:**

In general, I think this paper has achieved good performance as compared with SOTA methods. My major concerns are: 1) the gemoetry rendering branch is not clear; 2) whether smoothness plays major roles in achieving good geometry reconstruction; 3) the smoothness term is a bit tricky to tune.

---

> ### Author Response · Authors · 2022-11-18
> **Response to Reviewer VHoF.**
>
> **Question 1: Concerns about the assumption on color-geometry dependency.**
>
> **Answer 1:** Thank you for your constructive question. We have added a section in the supplementary material (Sec. D) to discuss this problem. Please refer to that part for the illustrative figures mentioned below.
>
> First, the assumption of color-geometry dependency is based on the idea of shape-from-shading, which has been proven effective in surface reconstruction by previous approaches ( Yariv et al., 2020, Yariv et al., 2021). This technique generally does more good than harm, while side effects do exist, like in the cases that the reviewer has mentioned. For example, we can observe obvious relief-like structures on a plane surface caused by the texture in Fig. S4.
>
> This is a common problem shared by most neural surface reconstruction methods (Oechsle et al., 2021; Yariv et al., 2020, Yariv et al., 2021; Wang et al., 2021). Nevertheless, this problem can be largely alleviated via multi-view consistency when the geometry and color fields are well-trained with enough input views. In Fig. S4, We observe that our method is better at overcoming the problem than previous state-of-the-art methods, being least affected by the textures. Fully addressing this side effect is out of the scope of this work and will be investigated in the future.
>
> **Question 2: How to render images just from the geometry branch.**
>
> **Answer 2:** Sorry for the confusion. As described in Sec. 5.2, the two branches of the dual color network both predict the color of a queried point based on the position, view direction, and other different conditions. We name the first network as $g_{geo}$ since it is mainly responsible for building the color-geometry dependency. The second branch $g_{feat}$ takes the local feature as inputs to enable more precise color learning, which in turn benefits geometry optimization.
>
> The predicted color values from the two branches, namely $c_0$ and $c$, can be used to render images via Equation (1), where $\hat{C}(r)$ rendered from $c$ is used for novel-view synthesis evaluations.
>
> **Question 3: How to tune the weighting parameters to stop over-smoothing?**
>
> **Answer 3:** Thank you for the thoughtful question. The hyper-parameters that control the strength of each smoothness term are empirically tuned, while the finalized values of them are quite robust to different scenes. We use the same set of hyperparameters in all the scenes in the DTU and BlendedMVS datasets, and most of the results are decent. We also extend the method to several new datasets, as shown in Fig. S5, Sec. E in the supplementary materials, where the weighting parameters still work well.
>
> **Question 4: What if we apply the smoothness constraints in the SDF + DVGO baseline method.**
>
> **Answer 4:** Thank you for your constructive suggestion. We conduct experiments to study the effect of smoothness constraints on the SDF + DVGO baseline, and we add a new section (Sec. C) in the supplementary material for discussion. Starting from the baseline model, we study the effect of the smoothness priors and their combinations proposed in the paper. Both the quantitative results and the qualitative results confirm that the smoothness constraints alone are not sufficient to produce accurate surface reconstruction, as apparent artifacts still exist (Fig. S3), and the CD is obviously higher than our method (Tab. R7). Meanwhile, adding strong smoothness constraints produces over-smoothed results, and it also significantly increases the time consumption of the baseline. Our method does not need to perform the heaviest smoothness constraints (e.g., Gaussian kernel for training) throughout the training due to the other key designs. More experimental results and analysis can be found in Sec. C.
>
> **Question 5: It would be better to also discuss the limitations of the proposed method.**
>
> **Answer 5:** Thank you for the constructive suggestion. We have added a new section (Sec. G) in the supplementary material to discuss the limitations of the proposed method.

---

### Author Response · Authors · 2022-11-18
**Response to all the reviewers.**

We sincerely thank all reviewers for their constructive suggestions and insightful comments on our work. We are encouraged that the reviewers find that our paper “has addressed a very critical issue” (Reviewer VHoF), “ is well motivated, sufficiently evaluated, and clear with its insights” (Reviewer 7aTz), “presents a well-performing and efficient method” (Reviewer 8Ao6);  the key designs including “dual color network and the hierarchical feature design are novel” (Reviewer kZqD).

We have posted responses to the comments of each reviewer, hoping our response can address your concerns.

We have also uploaded the revised paper and supplementary materials (the modifications are present in blue color), as summarized in the following. Please note Sec. B.3, C, D, E, F, and G in supplementary materials.

1. We add more ablation studies of smoothness designs on Voxurf. (Sec. B.3)

2. We evaluate the DVGO +NeuS with stronger smoothness designs. (Sec. C)

3.  We discuss the assumption of color-geometry dependency for surface reconstruction. (Sec. D)

4. We perform evaluations on other datasets. (Sec. E)

5. We further compare Voxurf with Point-NeRF. (Sec. F)

6. We discuss the limitations and further research directions of Voxurf (Sec. G)

7. We correct typos, citations, and writing suggestions.

Finally, we sincerely thank our reviewers and look forward to further discussions with you.

---

### Decision · Program_Chairs · 2023-01-20

**Decision:**

Accept: notable-top-25%

**Justification For Why Not Higher Score:**

The paper receives 2x marginally above the acceptance threshold and 2x accept, good paper.

**Justification For Why Not Lower Score:**

The reviews are mostly positive. All reviewers agree that the proposed work is significant, the experimental results are convincing. All the weaknesses mentioned in the initial reviews are addressed in the rebuttal.

**Metareview: Summary, Strengths And Weaknesses:**

This paper presents Voxurf, which is a hybrid voxel-grid- and MLP-based neural surface reconstruction method that is both fast and accurate. A volumetric representation of a voxel grid is used to speed up training, and the reconstruction accuracy is improved by a two-stage training process, a novel network that captures the relationship between color and geometry, and a more expressive SDF feature.

Strengths:
- Addressed a very critical issue when training Nerf, which is the extremely high computational cost.
- Achieved better performance on fast convergence as compared with NeuS, and more accurate geometry as compared with DVGO and the naive combination of DVGO and NeuS.
- Authors claim 20x improvement over fully implicit representation - which is not surprising (also see quality/misc below) but still impressive.
- An important strength of the paper is that ablations studies for all aspects are presented in the main paper and the supplement.

Weaknesses:
- Major concerns are: 1) the gemoetry rendering branch is not clear; 2) whether smoothness plays major roles in achieving good geometry reconstruction; 3) the smoothness term is a bit tricky to tune.
- Concerns about the scalability of this method to larger scenes, which is where the true benefit of the efficiency improvements would be particularly useful. Nevertheless, the authors' have provided a compelling case for the benefits of their method.

**Note From Pc:**

if the above contains the word "oral" or "spotlight" please see: "oral" presentation means -> notable-top-5% and "spotlight" means -> notable-top-25%. As stated in our emails, we are disassociating presentation type from AC recommendations